# Digitalisation in Craft Enterprises: Perceived Technostress, Readiness for Prevention and Countermeasures—A Qualitative Study

**DOI:** 10.3390/ijerph191811349

**Published:** 2022-09-09

**Authors:** Louisa Scheepers, Peter Angerer, Nico Dragano

**Affiliations:** 1Institute of Occupational, Social and Environmental Medicine, Centre for Health and Society, Faculty of Medicine, Heinrich Heine University, 40225 Düsseldorf, Germany; 2Institute of Medical Sociology, Centre for Health and Society, Medical Faculty, Heinrich Heine University, 40225 Düsseldorf, Germany

**Keywords:** technology, digital stress, work-stress, preventive intervention

## Abstract

Introduction: Digital technologies are increasingly used in the craft sector. Innovative technologies have several benefits for businesses but working with them can also induce negative feelings and stress. Data are lacking on perceived stress as well as the resulting strain and effects on health. However, information is needed about the principles of healthy working conditions in the context of digitalisation in the craft sector. To identify targeted starting points for preventive interventions to reduce technostress, we studied the perception of managers and employees of craft enterprises about digitalisation. Method: 26 guideline-based interviews were conducted with managers and employees in the craft sector to assess their understanding of digitalisation and the perceived effects on their health. The data analysis was based on a structured qualitative content analysis. Results: In the administrative area, digitalisation is perceived as simplifying work, as information, for example, becomes more easily accessible. The actual craft work, however, is perceived as more psychologically stressful, e.g., due to technologically more complex heating systems. Likewise, an increased stress experience is described in connection with constant accessibility and workplace surveillance. To reduce the stress experience, clear prevention wishes such as digital breaks or more participation in decisions about digital technologies are stated. Conclusion: Managers and employees in craft enterprises experience increasing psychological strain due to technostress. However, there is a willingness to take preventive interventions and a desire for more support in creating healthy working conditions in the context of digitalisation.

## 1. Introduction

The work environment is becoming increasingly digital. In particular, the use of information and communication technologies is on the rise. This is leading to fundamental changes in corporate culture, working conditions, work organization and communication [1,2,3,4,5]. On the one hand, digitalisation helps to increase efficiency, facilitates workflows, and represents immense potential for work organisations [1,4]. At the same time, digitalisation may create new workloads or exacerbate existing ones [6,7]. Employees have to deal with complex and new technologies and restructured work processes [1,2], which can result in stress reactions for users. These reactions can take the form of physical stress reactions such as the increase in the stress hormone cortisol, muscular tension pain, and headaches [8,9], or psychological stress reactions such as mental exhaustion, nervousness, and anxiety [2,3,10]. Work-related stress that persists over time can trigger negative health outcomes such as burnout, depression, or anxiety disorders [1,2]. The aforementioned stressful reactions associated with digital technologies are also referred to as “technostress” and arise from unsatisfactory work with technologies [11]. Creators of technostress (see Appendix A for further details) have been intensively researched for years [1,2,5,10,12,13]. Technostress has been found to be an important factor associated with job dissatisfaction, intentions to quit, declining workplace engagement, and increased absenteeism [1,2,5,10,14,15]. These findings are unsurprising, as according to Karasek’s model of work demand and control, an increase in work demands accompanied by a decrease in control by users over their own work can lead to an increase in work-related stress and result in long-term user illness [16,17]. However, stress can be buffered through various resources (e.g., social or organizational support, personal digital literacy) [16,18,19,20].

Despite an increasing number of studies on technostress being observed in large enterprises and office or production work, knowledge about the causes and severity of stress and its health consequences in the craft sector that mostly comprises micro- and small-sized enterprises (MSE) is almost completely lacking. In a mixed-method study on the topic of perceived software usability and usability-related stress in craft enterprises, we have already been able to show that usability problems are relevant as stressors in software use and are perceived as a burden by the employees [21]. However, as work in the craft sector has special features (working remote, close contact to clients, combination of physical work with engineering tasks, etc.), it is important to understand if technostress is comparable to the data sampled from large enterprises and different work sectors, or if technostress in the craft business differs from other businesses.

Twelve percent of all employees in Germany work in the craft sector, and in 2018 they generated nearly one-third (27%) of the total German economy [22], which makes craft enterprises an enormously important sector of the country’s economy and work environment. Digitalisation is not a trivial process for craft enterprises [23], as there is often a lack of time, financial resources, as well as human resources to systematically implement the digital transformation [22,24,25,26,27], and numerous differences between individual craft enterprises call for individual digital solutions [26,28]. However, the digitalisation offers new opportunities for personnel policy and economic management in craft enterprises themselves—for example, communication with customers and suppliers can take place in real time or warehousing can be reduced [28,29,30,31]. So far, however, digital technologies have been used in craft enterprises almost exclusively in the context of information and communication technology and rarely to improve work processes, because the potential is overlooked and knowledge for systematic introduction and reorganisation strategies is missing. Further problems in the implementation are the high costs of acquisition, the operational effort and the qualification of employees [24,28,31]. However, craft enterprises have already recognised the importance of digitalisation for the economy and competition, yet the implementation of digitalisation is highly dependent on the individual enterprises’ size and framework conditions [24,27].

The issue of mental health has received little attention, especially in MSEs as craft businesses [32], with only 27% of companies with less than ten employees investing in the health of their employees at all, and only 30% running regular risk assessments [33]. MSEs are difficult to reach for health actions, especially of (psychosocial) work design and workplace health promotion [33,34,35]. Barriers to implement a psychosocial risk assessment or preventive health actions are little flexibility in terms of time, financing, and human resources. In addition to this, employees often work in the technical field service in the craft sector [32,35], which makes it more difficult to introduce health-promoting measures. Likewise, it is becoming apparent that there is an information deficit on how to shape health in the workplace among managers [34]. However, clear and reliable evidence on the readiness for prevention or preventive measures in craft enterprises, especially in relation to stress perception in connection with digital technologies, is not available.

The aim of this study is to explore how managers and employees in craft enterprises (e.g., sanitary, heating, and air-conditioning business) perceive digitalisation and health to identify their stressors and stress reactions and to assess how ready they are for prevention or preventive measures. This is important to obtain missing information on occupational health risks in craft enterprises in connection with digitalisation and to uncover possible starting points to reduce technostress. For our study, we chose the sanitary, heating, and air-conditioning business because most of the employees in the craft sector work there [36] and they combine features of craft work with work on construction sites. Both require physical, mental, and social competence concerning customers, as well as an advanced degree of digitalisation. Due to the lack of previous studies, we conducted a qualitative interview study to answer the following research questions:Which creators of technostress and which health effects do managers and employees in the craft sector perceive in relation to digitalisation?To what extent are managers and employees in the craft sector aware of the connection between health and digitalisation?To what extent are managers and employees in the craft sector ready to take preventive measures to reduce technostress and what measures are conceivable?

## 2. Materials and Methods

### 2.1. Participants and Recruitment of Participants

The craft enterprises were identified via an Internet search and the association of the sanitary, heating, and air conditioning business in North Rhine-Westphalia in Germany. To contact and recruit participants, e-mails with project and study information were sent to 70 randomly selected craft enterprises. One week after sending these e-mails, we contacted the craft enterprises by telephone. In total, 18 companies agreed to participate. At least two years of professional experience in the sanitary, heating, and air conditioning sector were necessary for participation. 

The results are based on 26 semi-structured interviews (3 women) in 18 craft enterprises in North Rhine-Westphalia, Germany (see Table 1). The age of the participants ranged from 22 to 62 years with an average of 46 (SD = 8) years. Half of the participants stated that their occupation was manager of their craft business, 36% of the participants were technicians or plumbers and 14% were office workers. The size of the business ranged from 5 to 45 employees, with 33% having between 20 and 49 employees, 50% having between 10 and 19 employees and 17% having five to nine employees. During the interviews, participants reported working with specific craft enterprise software (e.g., to manage customer data, accounting, orders); e-mail programs; mobile applications (e.g., to identify errors in heating systems, thermographic cameras); video conferences; or social media (e.g., WhatsApp, Facebook).

### 2.2. Data Collection and Setting

The data were collected from February to March 2020. The interviews were conducted in craft enterprises during daily working hours to ensure a realistic setting [37]. The duration of the interviews was 45 to 90 min. All interviews were conducted by the first author. The content was recorded with digital audio recording devices. Interviews were conducted until data saturation was reached, i.e., to the point where the participants did not provide any new information but referred to information they had already given [38].

Each interview started with a short introduction informing the participants about the research focus. Once written informed consent was obtained, the interview started. The content and questions of the interview guide (Appendix A) focused on the participants’ professional situation, their role in the company, personal contacts, as well as previous experiences with digitalisation, the professional use of digital technologies, connection between digitalisation and health, health at work, and wishes towards digitalisation. The interview guide was jointly developed by the researchers based on information about craft enterprises [22,39] and the state of the literature on digitalisation and health [1,2,5,10,13]. The interview guide was piloted in December 2019 with two participants from the craft sector. No changes were necessary.

### 2.3. Data Analysis

The audio recordings were transcribed according to the extended criteria of Fuß & Karbach [40]. The interviews were analysed according to the qualitative content analysis of Kuckartz [41] using MAXQDA (version 2020, VERBI GmbH, Berlin, Germany) with a hermeneutic view of the data material. The first step of the content analysis consisted of marking particularly important text passages and collecting initial ideas for interpretation. These were then discussed in the research group (comprising the authors). The interview questions were transformed into deductive main categories (researchers’ suggestions, knowledge from the literature). In the analysis, these categories were further broken down into sub-categories through inductive category building. This is a common procedure in qualitative research, where a certain amount of prior knowledge or assumptions is already present [41]. A combination of deductive and inductive category building unites the following advantages of both processes: Deductive categories based on the interview questions and previous knowledge from the literature ensure that the data analysis stays focused on the research questions and connects the current results to previous knowledge. Thus, it can also prevent, to a certain extent, a bias of the data analysis and the formation of categories by the researchers compared to a purely inductive process. The inductive building of sub-categories, however, complements the deductive process and ensures that all the important topics and phenomena mentioned in the empirical data can be depicted [41]. In the final step of the analysis, the research-relevant answers were systematically summarised. To ensure the comprehensibility of the coding, a codebook was created in which the codes, their operationalisation, the coding rules, and anchor examples were listed (for examples, see Appendix A). 

Several strategies were used to ensure the trustworthiness and quality criteria of the qualitative analysis, e.g., transferability, reflexivity, or credibility [42]. A checklist was used to keep the interviews as consistent as possible and to ensure the transferability of the results. In addition, two forms of memos were used to analyse the interview data: an interview memo written immediately after the interview to reflect on and document the setting of the interview, and evaluation memos containing the researcher’s self-reflective thoughts during the interpretation process. Each step of the analysis was documented, e.g., the coding rules. The results were discussed within the research group. Additionally, the results were discussed with colleagues within a larger research project and preliminary results were presented at joint meetings to obtain feedback from experts and participants. To increase credibility, the transcripts were sent back to the participants to check the content.

### 2.4. Ethical Statement

Before the interviews started, all participants were informed verbally and in writing about the study procedure and data management. All participants signed a written informed consent form prior to participation. The data and transcripts were pseudonymised. The study was approved by the Ethics Committee of the Medical Faculty of Heinrich Heine University Düsseldorf (study no.: 2019-640).

## 3. Results

The results describe the perceived experiences, stresses and strains of managers and employees from the craft enterprises in relation to digitalisation. Data analyses revealed four major categories of answers to the question of technostress in craft enterprises: Four topics emerged: “Perceived connections between digitalisation and health”, “Perceived health effects due to digitalisation”, “Readiness for prevention” and “Company prevention measures”. We present the detailed results for these four categories in the following paragraphs (note that the quotes were translated from German to English).

### 3.1. To What Extent Are Managers and Employees in the Craft Sector Aware of the Connection between Health and Digitalisation?

When asked directly about technostress, almost half of the participants said they had not yet thought about it. Managers and employees described respective phenomena; however, on a more abstract level they could not make a connection between the stress and their health. The spontaneous statements of the participants tended to show that health in the craft sector is still primarily considered in physical terms. This way of thinking was more noticeable among the answers of the employees than among the managers. A specific awareness of the connections between digitalisation and health could not be inferred from the statements of managers and employees. 

“I have never thought about it before. […] I don’t know if health has anything to do with digitalisation. For that, digitalisation would at least have to be a cause first. Like getting sick”.(I_29B)

“Keeping healthy […] I don’t think that digitalisation can help much. […] I can’t say that yet, maybe my eyes will get worse at some point because of the screens […] and otherwise I don’t know how that should limit me”.(I_10B)

However, over the course of the interviews an examination of the connection between digitalisation and health was triggered. Consequently, associations between both aspects arose for most of the participants and almost frightened some of them. According to the participants, digitalisation should mean an increased quality of life. However, most participants doubted this and perceive digitalisation rather as a “double-edged sword” (I_16B). From an entrepreneurial point of view, the participants described digitalisation as a great opportunity for more professionalism and competitiveness, but on an individual level they saw health risks for the person who is confronted with new demands.

“[…] we have to be careful not to overtake ourselves with increasing digitalisation. So, from my point of view, digitalisation should be there to relieve people, to clear their heads. And not just to make people even faster or even more burdened […]”.(I_25B)

### 3.2. Which Creators of Technostress and Health Effects Do Managers and Employees in the Craft Sector Perceive in Relation to Digitalisation?

#### 3.2.1. Beneficial Effects of Digitalisation on Health

Two-thirds of the participants perceived digitalisation as a general facilitation for work organisation and processes, but not as an explicit promotor of their personal health. The perceived relief was nevertheless described by single participants as a kind of relaxation, relief or stress reduction. As reasons for the perceived facilitation of work, the participants counted time savings, more efficient work processes, location independence and transparent access to information—e.g., when researching and ordering spare parts or identifying errors in heating systems through mobile applications, craft enterprise software for managing customer data, billing and work orders, as well as specific software for the remote maintenance of heating systems.

“[…] in the past I had to have ordered [spare parts] by 5 o’clock p.m., otherwise the storekeeper was home. Today I order online and all night there are people loading the goods. Because it comes digitally to the warehouse worker on the joystick. That’s wonderful. Yes, of course it’s more stress-free”.(I_17B)

“Long live the app, […], when I think about it, when I did customer service in the past, I think I had 15 folders in the back of the car, for every producer a pool of error codes, now every heating system producer has their own app. You go in there and enter an error code, e.g., E19, then the app tells you directly what’s wrong. So that makes life mega easy for you”.(I_15B)

#### 3.2.2. Aspects of Digitalisation That Are a Health Risk

All the participants described several stress triggers on a personal level due to digitalisation and the use of new digital technologies, even if some of them were unwitting. Overall, the participants perceived digitalisation as fast-paced, leading to reduced rest periods and increased time pressure, as well as being associated with an increase in psychosomatic problems, stress and excessive demands. Compared to customers who would benefit from the improved and constant accessibility of craft enterprises, employees would not benefit from it since they would have to take all the messages and calls. According to the participants, digitalisation mainly serves to increase performance.

“You can’t get any rest at all. (…) And we have to be careful that we don’t lose a bit of humanity somewhere and then just say, only pressure, you have your orders, you have 17 min to get there and 17 min to get back and then we have to look, and if you need three minutes longer, then you lost. So, yes, I see a bit of a danger, I have nothing against clean processes, but we must not forget about humanity”.(I_25B)

##### Constant Accessibility

According to the perception of more than two thirds of the participants, constant accessibility and the associated effects are the “shortcoming” (I_22B) of digitalisation. With digitalisation, the expectation of customers to receive an immediate answer to emails, social media or text messages is growing. By using company smartphones work has become omnipresent, as emails and social media requests by customers can be retrieved and answered at any time. The same is true for calls. Therefore, participants also describe the smartphone as a “devil’s device” (I_10B) or as their “biggest enemy” (I_15B). According to the participants, there is a certain danger of working 24 h a day in order not to miss anything, which leads to not being able to switch off. According to their statements, managers and employees feel equally burdened by the dissolution of work boundaries. There is a feeling of being under constant pressure, which also leads to emotional overreactions, e.g., reacting irritably towards family members. Managers also describe that they always feel somewhat pressured by customers to call back immediately.

“The [constant availability] is really annoying. It can sometimes happen that it follows you around at night”.(I_14A)

“That I constantly have company-related thoughts in my head and can’t do much privately or slow down and come down, that you’re very often in contact with the company all the time”.(I_16B)

“What is definitely a disadvantage is to see the topic of e-mail again and again at any time, at any place, when Mrs. Müller writes something nasty on Sunday evening because she wasn’t satisfied with something on Friday, that eats into you a bit, so you really think to yourself,—‘No, I won’t look now’,—but you automatically look, let’s see, maybe there was something special or something, sometimes I would like to return to a different time when there was no mobile phone”.(I_15B)

##### Complexity of the Technology

The increasing complexity of the technologies in administration as well as in heating installation was stated as stressful and burdensome by almost all the participants. In particular, a lack of usability of the software solutions was critically described by the participants, since the use of software would make certain work processes more time-consuming and cumbersome. This is because, for example, unnecessary work steps in the installation process of a heating system cannot be skipped in the software. The craft enterprise software is very detailed in its functions and often only a small part of the functions and possibilities of the software is used. In addition, many software solutions are not self-explanatory according to the participants, and they are difficult to understand without additional effort or in-depth training. However, the use of hardware and software is now a prerequisite for many activities, according to the participants. According to the statements, the constant innovations in software solutions, the system errors that occur, and the limited compatibility of software programmes with each other represent a high burden, especially for older employees, which leads to a feeling of being overwhelmed. A detailed analysis focusing on the role of software usability in German craft enterprises is presented in Scheepers & Kaiser et al. [21].

“There are programs which you open and then you think—‘No, it doesn’t work at all’. Because we are not information scientists, we are craft worker!”.(I_15B)

“I get most annoyed when something about digitalisation, the PC or something else, something doesn’t work or I can’t do it, I get sick about it”.(I_19B)

##### Overload Due to Technology

More than half of the participants described a perceived increase in workload due to digitalisation. The type of work intensification was perceived differently by managers and employees, but both parties seemed to feel equally burdened, as expressed in the statements. 

Managers described the work intensification primarily in the administrative area, through an increasing volume of information, increased interruptions, and shortened processing times due to digital processing options. In the course of digitalisation, customers, for example, demand the faster processing of offers on the one hand, but also interrupt the work process with their queries via e-mails at the same time. For the preparation of offers and subsequent orders, the prices of the wholesalers must be kept in view to find the best prices. However, every wholesaler has a different ordering system on their website and a newsletter function, so it is difficult to keep track. According to the managers, digitalisation increases the pressure to work more effectively, which is sometimes compensated by multitasking to counteract the loss of control over their own work. In the end, however, the managers described an increase in mental stress. Excessive demands are made, which results in more mistakes. 

“[…] the office work is so much work for the size we have. Three people working full time in the office. In my eyes, this is becoming more and more”.(I_18A)

“When I started here, it was common that we needed four or five days to write an offer, that was no problem at all. Today, if the offer isn’t ready after two days, the next e-mail is ‘Where is my offer?’. So, people’s expectations have become higher in parallel with all this digitalisation that has progressed in recent years, and in my opinion society as a whole has also become much more stressful”.(I_28B)

Employees reported that the timing of the construction sites is becoming tighter, individual work steps are more time-consuming (e.g., explaining the technological handling of heating systems to customers), documentation is more work-consuming (e.g., every article from the warehouse has to be scanned and entered in the archive with the article number and linked to the order), and the processing of the digital worksheet is more complex than the analogue one. Just like the managers, the employees also reported an increase in the volume of information and more interruptions. For the employees, however, this is less due to customers but more due to queries from colleagues or managers, e.g., via social media, telephone or e-mails. The employees also described that they try to counteract the work overload and information overload through multitasking. This also leads to stress among employees, which manifests itself in a loss of concentration and increased errors. 

“[…] with the digital gimmicks, more work is pushed onto me. If I perhaps previously filled out my work sheet in handwriting and used the materials in handwriting, now I have to do it digitally, which means I now have to pick out the right article, deposit it there, write the text sensibly and that can of course also trigger stress for some people, of course, because it is more work at that moment”.(I_11B)

“Well, I counted once, I had 35 calls by eleven o’clock, while I was screwing, fortunately all routine work, I was giving technical advice or answering questions for colleagues, […]”.(I_21B)

##### Workplace Surveillance by Employer and Customers

More than half of the interviewed employees described stress due to surveillance by the employer as well as by the customers, whereby the perceived pressure of surveillance by the employer seemed to be more pronounced. The feeling of becoming a “transparent employee” (I_16B) was described, as GPS tracking systems in work vehicles, digital driver’s logs, and working time recording with location determination give the craft enterprise more and more possibilities to draw statistical comparisons between employees. This indirectly places workers under increasing pressure to improve their performance, which was associated with feelings of distrust by the employees. Some employees also felt that this was an invasion of their privacy. The managers see the collected data more as an aid for the employees to document their work and to be able to justify themselves to customers, e.g., in the case of critical customers regarding their pay.

“The fact that you are always confronted with how long it took me, why did it take me so long or was someone else faster—for the employer it is then of course easy to evaluate these things, to somehow pull them up as statistics and to see who is how efficient. […] based on the metadata from an app, you can also somehow determine a lot, how long he took at lunchtime, how long he needed […]”.(I_10B)

Workplace surveillance by the customer is stressful for managers and employees alike. Customers are much more informed through Google, YouTube, MyHammer, etc. This increases the pressure on the managers to have immediate professional knowledge. On the phone, customers often already mention what is broken in their heating system. On site, the administrative process is closely followed, from the information on the invoice, to the installation or repair. As a result, the participants experience a high level of mental pressure, which leads to a feeling of tension and sometimes frustration (noticeable in the participants’ expressions). 

“‘Why is it taking you so long? On the Internet, he was faster than you’. Or the customers know better. They call and say, this and that is broken in my heating. That’s the other side [of digitalisation], customers inform themselves on websites or watch some YouTube videos”.(I_17B)

### 3.3. To What Extent Are Managers and Employees in the Craft Sector Ready to Take Preventive Measures to Reduce Technostress, and What Measures Are Conceivable?

#### 3.3.1. Willingness of Managers and Employees to Take Preventive Measures

Managers and employees of craft enterprises are equally willing to take preventive measures regarding health. There is an awareness that the work in the craft enterprise sector is a physically demanding profession, and that physical work is an essential part of the business (e.g., installing a heavy boiler), no matter how much digitalisation is involved. In addition, the employees are aware of the dangers that the job entails, and they know the personal protective measures. However, the participants’ statements suggest that they might not know how to deal with the possible health risks associated with digitalisation. 

“[…] Work has to be done, and it is unavoidable that someone has to do it, so no matter how much digitalisation I put into it, in the end I still have to turn the screw by hand as a rule”.(I_10B)

The statements of the participants, especially of the managers, also show that the willingness to engage in preventive health measures is influenced by external factors. The measures should be cost-effective and take as little time as possible. The time factor outweighs the investments that may be necessary in terms of priority. There is a willingness on the part of employees to become involved in new issues and, if necessary, to learn something new. Managers and employees would like to have professional help with prevention work, as they know that there is a lack of personnel competence in craft enterprises for this—for example, in the form of contact persons at the occupation cooperative or personal advice and support at the workplace. Managers in particular reported a need for more awareness for the issue of health in everyday working life. However, according to the managers, the craft enterprises are also prepared to become active themselves and to organise group meetings within the workforce to address health issues. Managers would also like to involve employees more in the process of health prevention. Several managers described that they are happy to receive suggestions from employees. However, it was also clearly stated that in everyday life, the focus is naturally more on the economic process than on health—health comes more into focus when accidents occur. 

“Health as a topic, but of course I have to get the employees on board. Actually, first and foremost, because they are the ones who are really affected—communication, communication, communication. […] So there are certainly starting points”.(I_25B)

#### 3.3.2. Measures to Reduce Technostress Triggers in the Craft Enterprise

##### Participation in the Digitalisation Process 

Participation in the implementation of new digital technologies is important to half of the employees interviewed, as this motivates them to help shape the changes, to deal with any difficulties, and to help them recognise personal benefits. Some workers said that they had been asked for feedback after the introduction of new technologies, but most employees reported that new technologies had only been selected and introduced by management without them being involved. According to the employees, this leads to overload when the workload otherwise remains the same, which results in frustration and decreased job satisfaction. 

“Yes, often you are simply confronted with these things and hear ‘Here, it’s new now, you can try it out’. When you test it, you notice if it gets better or easier. Everything they no longer have to do in the office, someone else has to do and the workload remains the same. Documentation and organisation and so on, you just have to look, because if I as a technician have to do more [e.g., documentation], then I have less time for other things”.(I_10B)

##### Clear Communication Strategies

According to the managers and employees, it is important to prevent the removal of boundaries by technology by defining clear communication strategies in the craft enterprise, in order to create “digital breaks” (I_15B). Ideas were described such as the definition of availability, the type of communication channels (via email or messenger), checking emails at defined times, the use of an answering system or telephone secretary, or machines that automatically shut down during regular breaks. To protect the privacy of employees, it was also essential that employees’ internal work-related mobile phone numbers were not passed on to customers. 

“[…] when everything is more digitalised, it tends to increase, the stress increases from this accessibility and so on. […] maybe that you just determine when you discuss things or something. I don’t know”.(I_20B)

“[…] that you give the employees the opportunity to have a moment to switch off. Well, not this constant firing up with emails, I think that would be important. To have quiet times from all this [digitalisation]”.(I_14B)

##### Application Training 

In the interviews, the participants described that they had deliberately chosen a practical and craft profession. Due to digitalisation, they would now be confronted more and more with new technologies for which they lack the IT knowledge. In order to reduce the associated digital stress due to the increasing complexity of technologies, more internal and external training and information opportunities are necessary, according to the participants. In addition, the participants said that the issue should be increasingly integrated into the training.

“I would say that the training position is not given in such a way that you are better trained in hardware, programs or so”.(I_19A)

##### Support Systems

The participants stated that they usually did not have a specific contact person for IT problems in the craft enterprise and that they had to spend a lot of time researching hotlines for help. In addition, the manufacturer hotlines are difficult to reach, and, in the end, there is often no satisfactory answer that helps to solve the problem. Therefore, the wish was expressed several times to provide an in-house contact person for IT problems and to create lists for external contact persons and make them accessible to everyone. 

“Disaster, nowadays you only have hotlines everywhere. And in the end, when you get someone, they don’t have a clue because it’s just one person taking a phone call. If you’re lucky, unless it’s just computers”.(I_17A)

“[…] we also need an IT consultant who can explain the whole thing [software] to us. It would be nice if we could work hand in hand like that”.(I_19A)

## 4. Discussion

In the present study, we investigated the perceived health effects and stress reactions of managers and employees in craft enterprises due to digitalisation. Moreover, we identified possible starting points for effective health-promoting measures in dealing with digital work-related stress in the craft enterprise sector.

Digitalisation is leading to profound changes in work organisation [1,2,3,5], which managers and employees must deal with and which can lead to work-related stress [1,2]. Our qualitative findings are consistent with this state of knowledge but also provide a new insight into the emergence of technostress in the craft sector. Various perceived health effects were described by participants, revealing a range of new or intensifying psychological stressors. However, most of the participants had not yet experienced a deeper engagement with the issue, so there remains no specific awareness of the connection between digitalisation and health. However, there is a desire and willingness for prevention regarding digital-related stress in the workplace.

The participants positively emphasised the relief provided by digitalisation at the administrative level, as information is more easily accessible and managed. Time is saved and economic efficiency increases as a result. The changes at work organisation level create new opportunities for design leeway [28,29,30,31], but due to a lack of knowledge about the digital transformation process and resources in the craft enterprise, these have not yet been exhausted [23,24,27,28]. This is also reflected in the interview results, in the form of a desire for more support in the implementation of digitalisation, as well as application training.

On a personal level, the participants described clear triggers for technostress, even if they were not yet specifically aware of them in relation to health. The stressors described for technostress, as already known from the literature, also lead to stress reactions such as excessive demands and are accompanied by the perception of increased psychosomatic complaints [1,2]. However, the extent of the descriptions on the dissolution of boundaries between private and professional life is surprising, as this is initially not to be expected for work in the craft sector that is carried out on site for the customer. However, the participants described that due to digital possibilities, e-mails and software, data (e.g., customer and heating system data) can be accessed at any time and customers use any contact possibilities after work. Consequently, the participants reported a feeling of the omnipresence of work, which makes it more difficult to switch off from work, shortening rest periods. Furthermore, our results show that the participants also face increasingly complex, fast-moving and error-prone technologies in heating construction, which has already been illustrated by other studies [1,2]. According to our findings, the usability of software has an important influence on the perception of stress, as it does not sufficiently support the workflow of the participants. The fact that good usability makes work easier and, thus, also reduces digitally-induced work stress has already been confirmed [43], but it has not yet been sufficiently taken into account in SMEs and craft enterprises [21,44]. It is also known from research on technostress that digitalisation forces the users of the technology to work faster and longer than recommended [2,5]. This is also reflected in our findings, but the way that techno-overload is perceived differs between managers and employees. Managers perceived a work overload especially in the administrative area, whereas employees mainly described a tighter timing of work at construction sites and higher documentation efforts. However, both parties described a higher volume of information and increased interruptions, although for different reasons. The participants try to compensate for this with multitasking, which leads to more errors, as research also shows [45]. Furthermore, the participants described workplace surveillance by customers as a trigger for technostress that has not yet been addressed in the literature. Customers are more informed by the technologies about individual work steps and time sequences, which are then matched with the technician’s performance. The classic workplace monitoring, which is named as a stress trigger, is the control of work performance, work location, and working hours by the employer [1,13]. With increasing digitalisation, employees experience themselves more and more as transparent, due to possible efficiency statistics or GPS tracking. 

As a result of the perceived stressors and the associated technostress, managers and employees feel overwhelmed. Overstrain and stress lead to mistrust and a loss of control, while at the same time the demands on work increase due to digitalisation. In general, digitisation allows work processes to be tracked more intensively, which means that more workplace surveillance is taking place and more pressure is being built up on employees. Customers expect ever shorter processing and response times on the part of managers, which entails the risk of ever more acceleration and pressure to perform. Ultimately, according to Karasek’s work demand and control model [16], these factors lead to high job stress, and it has already been researched that technostress can be linked to it [1,45]. The outcome is a decrease in job satisfaction and an increased risk of depression and cardiovascular disease due to technostress [10,15,17,18,45,46].

In order to take the harmful effects of digitalisation into account, according to Georg & Guhlemann [6] and Janda & Guhlemann [7], it is crucial to counteract the new and changed workloads. This requires a certain degree of readiness for prevention on the part of the companies, but many managers in MSEs do not see any need for action so far, as the risks and hazards for mental illness in particular do not seem to be known [32,35,47]. Our results positively indicate a general willingness to prevent technostress on the part of managers and employees. Among employees, there is still a lack of knowledge in dealing with the health risks associated with digitalisation. Occupational health and safety and health promotion should, therefore, consider addressing this lack, providing information on this issue to the employees and strengthening the personal coping capacities [6,7]. Managers in particular know that there is a need for more awareness of the topic of health, inter alia, in connection with digitalisation, which is also referred to by Amler et al. [34] when considering statutory health protection. Managers also show a clear willingness to involve employees more in the process of health prevention in the company and to allow them to participate in it.

The participation of end-users in decision-making processes when introducing new technologies or implementing them, e.g., in the form of employee surveys, team discussions on software selection and the use of test phases, is shown to be an important factor in reducing digital-related work stress [18,19,46]. Likewise, the employees in our study signalled a willingness to learn new things. However, it is also clear from the results that there is a lack of knowledge on how to prevent technostress and that help from external sources is needed for a systematic approach. For the managers and employees alike, the willingness to take preventive measures is also linked to how time-consuming and cost-intensive the respective measures are. However, the higher the resources required, the more likely the measures are to be deferred. Although the general wishes of companies regarding the design of preventive measures are known [32], further specific research is needed on the subject of digitalisation. Initial recommendations for action, based on the results of this study and other research project results, can be found under www.handwerkwirddigital.de (accessed on 10 August 2022). 

The participants further see a potential to reduce digital stress in making clear communication agreements such as establishing availability or checking emails proactively [1], in order to create digital breaks. Another key factor contributing to increased user satisfaction is the expansion of IT knowledge [18,19]. The participants would like this to be available in the form of more specific and individualised training and information opportunities. Another stress-reducing factor is the appropriate distribution of workload [1,19], which, according to the participants, has not yet been taken into account at the organisational level in the craft enterprise. Rather, the achieved free spaces are filled again by digitalisation (e.g., tighter time frames for construction sites). 

Overall, the results show that the perceived stressors related to digitalisation and the resulting strains are not only a phenomenon for professions such as managers [48], but also for the craft sector. The digitalisation of work requires health-promoting or preventive measures to reduce digital-related work stress. To this end, employees should be involved in decision-making processes and clear communication strategies should be implemented to counteract the dissolution of boundaries between work and private life. In addition, legal protective measures may be needed, e.g., to regulate digital workplace surveillance. However, further research in the craft sector is needed to quantify the aspects found here in a larger sample of craft workers, in order to further differentiate the various needs of managers and employees to ultimately be able to make evaluated prevention recommendations. 

## 5. Strength and Limitations

The strength of the study lies in providing, for the first time, a comprehensive insight into the perceived stress caused by digitalisation in the craft enterprise sector, as well as initial indications for preventive measures to reduce technostress there. Qualitative research offers the opportunity to provide comprehensive information on a so-far little-known topic and to obtain an initial overview of the mood. The results presented here refer to the sanitary, heating, and air conditioning business and may be different in other craft sectors, as well as in MSEs and SMEs. No claim is made to generalise the results. As a limitation, a certain selection bias cannot be ruled out, since on the one hand, it could be argued that only those managers and employees who have already dealt with the topic of digitalisation in craft enterprises participated and have, thus, developed a certain awareness of the associated difficulties. However, based on the results, which show that there has not yet been a specific awareness of the connection between digitalisation and health, this is unlikely. On the other hand, it can be argued that other managers and employees may not have participated due to the additional workload and complexity created by digitalisation. Further quantitative research is necessary to verify the representativeness of the presented results. 

## 6. Conclusions

The participants’ perceptions of the effects of digitalisation on health largely coincide with the triggers for technostress described in the literature. A new aspect is the perceived surveillance of the workplace by the customer. Technostress is, thus, also an increasing factor for psychological stress in the workplace in craft enterprises, which should be taken into account in occupational health management and health prevention. A lack of specific awareness of the fact that the digitalisation of work processes can have health effects becomes obvious by the fact that new digital technologies are still selected and introduced by the management level, often without the participation of the employees. However, there is a willingness on the part of both managers and employees to tackle and implement preventive measures to reduce technostress. The desire for more support for the implementation of the digital transformation process and individual prevention offers is particularly clear.

## Figures and Tables

**Table 1 ijerph-19-11349-t001:** Overview of participants and craft enterprises.

Company-Nr.	Digitalisation Level of the Company *	Number of Employees	Int.-Nr	Sex ^#^	Age	Role	Qualification	Duration of Employment in Years
**1**	highly	25	1	M	32	fitter/electrician	vocational training	3
			2	M	29	customer service fitter/foreman	masters	17
**2**	highly	45	3	M	22	heating installer, fitter	vocational training	6
			4	M	32	customer service fitter	vocational training	6
**3**	moderately	5	5	M	60	managing director	masters	27
				F	56	commercial clerk	vocational training	27
**4**	moderately	9	6	M	40	managing director	masters	15
**5**	moderately	10	7	M	55	managing director	study	19
**6**	moderately	6	8	F	52	commercial clerk	vocational training	11
				M	61	managing director	masters	32
**7**	moderately	10	9	M	21	commercial clerk, fitter	vocational training	6
				M	55	managing director	masters	20
**8**	moderately	10	10	M	54	commercial clerk	vocational training	15
				M	54	managing director	masters	29
**9**	moderately	35	11	M	23	customer service fitter	vocational training	*
			12	M	58	customer service fitter	vocational training	32
**10**	highly	40	13	M	46	site management, safety specialist, fitter	masters	28
**11**	moderately	13	14	M	48	customer service fitter	vocational training	*
**12**	highly	14	15	M	54	managing director	study	18
**13**	highly	16	16	M	62	managing director	masters	48
**14**	moderately	16	17	M	54	managing director	masters	25
**15**	moderately	15	18	M	41	managing director	study	22
**16**	moderately	30	19	M	30	managing director	study	10
**17**	moderately	36	20	M	58	managing director	study	23
**18**	highly	13	21	F	49	managing director	study	23
				M	43	customer service fitter	vocational training	2

* Participants’ self-declaration; ^#^ M = male; F = female.

## Data Availability

https://osf.io/ns7e5/ (accessed on 10 August 2022).

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
