# Peer review of "Digitalisation in Craft Enterprises: Perceived Technostress, Readiness for Prevention and Countermeasures—A Qualitative Study"

_ijerph, 2022, doi:10.3390/ijerph191811349_

Round 1
Reviewer 1 Report
Dear authors,
Thank you for the opportunity to engage with your work.
The project of technostress in craftwork is an interesting and timely contribution to the wider literature set you engage with. My comments are therefore, limited to some areas where I feel the contributions can be strengthened, particularly in the analysis/discussion section.
Overall, I found the article to be well structured, coherent, and well written. The various sections are all in place and the outcomes of the study are clear.
Data analysis
- Might you say something more about the hybrid process of your data analysis? Here I’m thinking of some sentences that tell the reader what hybrid analysis is, how it works, and what it is working up against, that is, what are the shortfalls within non-hybrid processes that makes it necessary/useful.
Results
- I find that the language of emergence within the results section to be an interesting choice. These subheadings seem very much tied to the research questions you outline, which is fine, it’s just that it strikes me as less about the ‘emergence’ of categories, and more about the research design setup.
- You write that the interviewees did not associate digitalization with questions of health, yet they list a series of stresses and strains that digitalization has brought on. These stresses and strains are clearly articulated as questions of personal health, and workplace thriving (or lack of it). So, in what sense do they not make this connection? I, of course, don’t know what questions you asked, but I’m guessing (so bear with me here) that one possible explanation could be the more abstract nature of questions about digitalizations and health. If these questions were put to participants at the start, then it’s quite possible they don’t acknowledge the connection. But the examples they give indicates a clear sense of digitalization and health. So, maybe, again, this is more about research design, questions, their order in the interview, and how much weight you attach to specific questions. Again, a short paragraph clarifying this would be welcome.
- The one area where I really think you can strengthen the article is by ‘daring’ to say more analytically and conceptually. You give us a nice and tight accounting of what was said, but you could say more about the connectivities around what was said. In other words, might you be able to offer some conceptualizations of what is going on? Here are a couple of examples for you to think about:
o P8, 3.2.2.3, interview citation: “So, people's expectations have become higher in parallel with all this digitalisation that has progressed in recent years, and in my opinion society as a whole has also become much more stressful."
o So, could you offer us some way of thinking about this relationship between expectations and acceleration?
o Or on p9, 3.2.2.4, what might we be able to say about the interesting connection between higher expectations and surveillance, how might we think about this in terms of digitalization and the workplace?
Discussion
- I would also enjoy seeing a bit more work within the discussion section, particularly on prevention and participation. My sense is that there is a slight management bias here, and that you could do more to talk about the structural conditions through which these categories are being articulated.
o You say, on p12, that “Our results positively indicate a willingness to prevent on the part of managers and employees. Managers in particular are aware that there is a need for more awareness of the topic of health inter alia in connection with digitalisation” and again on the same page “For the participants, the willingness to take preventive measures is also linked to how time-consuming and cost-intensive the respective measures are.”
o So in the first citation we learn that managers are aware that there is a need for more awareness. I’m wondering if you could say more than this? In the second, the barriers are clearly quite high, which points towards managers caring about awareness and prevention as long as it doesn’t cost the firm time and money (in the traditional sense). To be honest, one classic organizational response to problems in the workplace is deferral. Might you be buying in to this a little bit here?
o Also, in terms of participation. Again, on p12 you say “Participation of end-users in decision-making processes when introducing new technologies or implementing them is shown to be an important factor in reducing digital related work stress”
o It’s hard to disagree with this. But what types and forms of meaningful participation are in play or are something that could benefit workers?
o On p13, your closing remarks say this “Overall, the results show that the perceived stressors related to digitalisation and the resulting strains are not only a phenomenon for professions such as managers, but also for the craft sector.”
o Again, I don’t think this will surprise anyone. I appreciate that this is an early study in this sector and that more study is needed, but what might you be able to say through this study about the ongoing encroachment of digitalization into the workplace in terms of, for example, participation, prevention, surveillance, and expectations. Again, my push to you guys would be to do a little more conceptual work.
Author Response
Dear Reviewer,
we would like to thank you for the effort you took to evaluate our study. Your comments really helped to amend the quality of the paper. We have read all comments carefully and a detailed point-by-point response to your comments can be found on the following pages. Additionally, we have highlighted changes in the manuscript using track change.
General rating
Thank you for the opportunity to engage with your work.
The project of technostress in craftwork is an interesting and timely contribution to the wider literature set you engage with. My comments are therefore, limited to some areas where I feel the contributions can be strengthened, particularly in the analysis/discussion section.
Overall, I found the article to be well structured, coherent, and well written. The various sections are all in place and the outcomes of the study are clear.
- We are very pleased with your positive and appreciative words and would like to thank you once again for agreeing to carefully review this manuscript. Below you will find our responses to your specific comments and queries, point by point.
Data Analysis
Might you say something more about the hybrid process of your data analysis? Here I’m thinking of some sentences that tell the reader what hybrid analysis is, how it works, and what it is working up against, that is, what are the shortfalls within non-hybrid processes that makes it necessary/useful.
- Thank you for the question and the comment. By “hybrid process” we refer to the combination of deductive and inductive processes during category building. However, we now realize that this wording may be misleading. Hence, we clarified the wording in the manuscript and additionally added a clarification of the advantages of a combination of deductive and inductive processes during data analysis. These changes can be found in the data analysis section (page 5, lines 160ff.).
The respective text passage reads as follows: “The interview questions were transformed into deductive main categories (researchers' suggestions, knowledge from the literature). In the analysis, these categories were further broken down into sub-categories through inductive category building. This is a common procedure in qualitative research, where a certain amount of prior knowledge or assumptions is already present (41). A combination of deductive and inductive category building unites the following advantages of both processes: Deductive categories based on the interview questions and previous knowledge from literature ensure that the data analysis stays focused on the research questions and connects the current results to previous knowledge. Thus, it can also prevent, to a certain extent, a bias of the data analysis and formation of categories by the researchers compared to a purely inductive process. The inductive building of sub-categories, however, complements the deductive process and ensures that all important topics and phenomena mentioned in the empirical data can be depicted (41).”
Results
I find that the language of emergence within the results section to be an interesting choice. These subheadings seem very much tied to the research questions you outline, which is fine, it’s just that it strikes me as less about the ‘emergence’ of categories, and more about the research design setup.
- The structure of the results is based on the research questions and the topics mentioned by the interviewees in the empirical data alike, as the categories emerged by a combination of deductive and inductive category building as explained in the data analysis section. Due to the deductive category formation within the framework of the analysis process, there is naturally a certain theory-based structure to the research questions. We included a short explanation at the beginning of the results section on how the four major categories emerged (page 6, lines 197ff.).
The respective text passage reads as follows: “The results describe the perceived experiences, stresses, and strains of managers and employees from the craft enterprises in relation to digitalisation. Data analyses revealed four major categories of answers to the question of technostress in craft enterprises: Four topics emerged: "Perceived connections between digitalisation and health", "Perceived health effects due to digitalisation", "Readiness for prevention" and "Company prevention measures". We present the detailed results for these four categories in the following paragraphs (note, that the quotes were translated from German to English).”
You write that the interviewees did not associate digitalization with questions of health, yet they list a series of stresses and strains that digitalization has brought on. These stresses and strains are clearly articulated as questions of personal health, and workplace thriving (or lack of it). So, in what sense do they not make this connection? I, of course, don’t know what questions you asked, but I’m guessing (so bear with me here) that one possible explanation could be the more abstract nature of questions about digitalizations and health. If these questions were put to participants at the start, then it’s quite possible they don’t acknowledge the connection. But the examples they give indicates a clear sense of digitalization and health. So, maybe, again, this is more about research design, questions, their order in the interview, and how much weight you attach to specific questions. Again, a short paragraph clarifying this would be welcome.
- In the first part of the interview, participants were briefly asked about their work situation, followed by questions about digitalisation (e. g. ‘What do you understand under digitalisation?’ or ‘At what points in your daily work do you come into contact with digital media or digital work devices?’). In the main part of the interview, the participants were asked about digitalisation and health (What connections do you see between your work, digitalisation, and your health?). The entire questionnaire can be found under https://osf.io/ns7e5/?view_only=ad5650a636774ab598fdc2fd6c40a2e9.
When asked directly, participants first failed to report being aware of connections between their health and digitalisation. In their further answers, however, participants have indeed an understanding of the particular strains associations with digitalisation. Hence, the were not aware that this also constitutes an association between digitalisation and health. We have tried to show this more clearly in the article (page 6, lines 204ff.).
The respective text passage reads as follows: “Although when asked directly about technostress, almost half of the participants said they had not yet thought about it. Managers and employees describe respective phenomena, however on a more abstract level they cannot make a connection between the stress and their health. The spontaneous statements of the participants tended to show that health in the craft sector is still primarily considered in physical terms. This way of thinking is more noticeable among the answers of the employees than among the managers. A specific awareness of the connections between digitalisation and health cannot be inferred from the statements of managers and employees.
"I have never thought about it before. [...] I don't know if health has anything to do with digitalisation. For that, digitalisation would at least have to be a cause first. Like getting sick." (I_29B)
"Keeping healthy [...] I don't think that digitalisation can help much. [...] I can't say that yet, maybe my eyes will get worse at some point because of the screens [...] and otherwise I don't know how that should limit me" (I_10B).
However, over the course of the interviews an examination of the connection between digitalisation and health was triggered. Consequently, associations between both aspects arose for most of the participants and almost frightened some of them. According to the participants, digitalisation should mean increased quality of life.”
The one area where I really think you can strengthen the article is by ‘daring’ to say more analytically and conceptually. You give us a nice and tight accounting of what was said, but you could say more about the connectivities around what was said. In other words, might you be able to offer some conceptualizations of what is going on? Here are a couple of examples for you to think about:
o P8, 3.2.2.3, interview citation: “So, people's expectations have become higher in parallel with all this digitalisation that has progressed in recent years, and in my opinion society as a whole has also become much more stressful."
o So, could you offer us some way of thinking about this relationship between expectations and acceleration?
o Or on p9, 3.2.2.4, what might we be able to say about the interesting connection between higher expectations and surveillance, how might we think about this in terms of digitalization and the workplace?
- Thank you very much for pointing out this important limitation! Yet, including conceptual explanations in the results section of a qualitative study seem to be risky to us, as this already involves a deeper interpretation of the results and goes beyond their descriptive presentation. Nevertheless, we have tried to comply with your suggestion by adding some thoughts about this issue in the discussion section (page 13, lines 544ff. and 593ff.).
The respective text passage reads as follows: (lines 544ff ) “In general, digitisation allows work processes to be tracked more intensively, which means that more workplace surveillance is taking place and more pressure is being built up on employees. Customers expect ever shorter processing and response times on the part of managers, which entails the risk of ever more acceleration and pressure to perform.”
(lines 593ff.) “The digitalisation of work requires health-promoting or preventive measures to reduce digital-related work stress. To this end, employees should be involved in decision-making processes and clear communication strategies should be implemented to counteract the dissolution of boundaries between work and private life. In addition, legal protective measures may be needed, e. g. to regulate digital workplace surveillance.”
Discussion
- I would also enjoy seeing a bit more work within the discussion section, particularly on prevention and participation. My sense is that there is a slight management bias here, and that you could do more to talk about the structural conditions through which these categories are being articulated.
o You say, on p12, that “Our results positively indicate a willingness to prevent on the part of managers and employees. Managers in particular are aware that there is a need for more awareness of the topic of health inter alia in connection with digitalisation” and again on the same page “For the participants, the willingness to take preventive measures is also linked to how time-consuming and cost-intensive the respective measures are.”
o So in the first citation we learn that managers are aware that there is a need for more awareness. I’m wondering if you could say more than this? In the second, the barriers are clearly quite high, which points towards managers caring about awareness and prevention as long as it doesn’t cost the firm time and money (in the traditional sense). To be honest, one classic organizational response to problems in the workplace is deferral. Might you be buying in to this a little bit here?
- Thank you for pointing that out. It seems indeed to be the case, that managers are aware that there is a need to build more awareness for the connections between digitalisation and health for the employees and to take preventive measures to ensure physical and psychological health. However, this extent of willingness is closely related to the resources that are necessary to implement those measures. As suggested, we expanded our thoughts on how this might be related to deferral in the manuscript. (page 13, lines 558ff.)
The respective text passage reads as follows: “Our results positively indicate a general willingness to prevent on the part of managers and employees. Among employees, there is still a lack of knowledge in dealing with the health risks associated with digitalisation. Occupational health and safety and health promotion should therefore consider to address this lack and to provide information on this issue to the employees and to strengthen the personal comping capacities (6, 7). Managers in particular know that there is a need for more awareness of the topic of health, inter alia, in connection with digitalisation, which is also referred to by Amler et al. (34) when considering statutory health protection. Managers also show a clear willingness to involve employees more in the process of health prevention in the company and to let them participate in it.
Participation of end-users in decision-making processes when introducing new technologies or implementing them, e. g. in the form of employee surveys, team discussions on software selection and use or test phases, is shown to be an important factor in reducing digital-related work stress (19, 18, 46). Likewise, employees in our study signal a willingness to learn new things. However, it is also clear from the results that there is a lack of knowledge on how to prevent technostress and that help from outside is needed for a systematic approach. For the managers and employees alike, the willingness to take preventive measures is also linked to how time-consuming and cost-intensive the respective measures are. However, the higher the resources required, the more likely the measures are to be deferred.”
o Also, in terms of participation. Again, on p12 you say “Participation of end-users in decision-making processes when introducing new technologies or implementing them is shown to be an important factor in reducing digital related work stress”
o It’s hard to disagree with this. But what types and forms of meaningful participation are in play or are something that could benefit workers?
- Thank you for this advice. We have now added some examples for successful measures from the literature (page 13, line 568f.)
The respective sentence now reads: “Participation of end-users in decision-making processes when introducing new technologies or implementing them, e. g. in the form of employee surveys, team discussions on software selection and use or test phases, is shown to be an important factor in reducing digital-related work stress (18, 19, 45)”.
o On p13, your closing remarks say this “Overall, the results show that the perceived stressors related to digitalisation and the resulting strains are not only a phenomenon for professions such as managers, but also for the craft sector.”
o Again, I don’t think this will surprise anyone. I appreciate that this is an early study in this sector and that more study is needed, but what might you be able to say through this study about the ongoing encroachment of digitalization into the workplace in terms of, for example, participation, prevention, surveillance, and expectations. Again, my push to you guys would be to do a little more conceptual work.
Again, we agree and thank you for this advice. In order to make this clearer we have added the following sentences to the discussion section (page 13, lines 593ff.): “The digitalisation of work requires health-promoting or preventive measures to reduce digital-related work stress. To this end, employees should be involved in decision-making processes and clear communication strategies should be implemented to counteract the dissolution of boundaries between work and private life. In addition, legal protective measures may be needed, e. g. to regulate digital workplace surveillance.”
Sincerely,
Louisa Scheepers
Reviewer 2 Report
This is an excellent study. A few minor punctuation and word usage errors that can be fixed in editing. Otherwise, very interesting read throughout. The direct comments of the participants were very revealing. I find many of their concerns are mine as well in the teaching field. Well done.
Author Response
Dear Reviewer,
we would like to thank you for the effort you took to evaluate our study. Please find an answer to your comment below.
This is an excellent study. Otherwise, very interesting read throughout. The direct comments of the participants were very revealing. I find many of their concerns are mine as well in the teaching field. Well done.
- We are very pleased with your positive and appreciative words and would like to thank you once again for agreeing to review this manuscript.
A few minor punctuation and word usage errors that can be fixed in editing.
- Thank you for pointing that out. We have transferred the manuscript to a proof reader who has carefully checked the paper and revised several errors (corresponding changes are marked in the revised manuscript).
Sincerely,
Louisa Scheepers
Reviewer 3 Report
>Section 2.3. Data analysis, line 164 (p. 5 of 15): This study adopts a qualitative research design; discuss how you are going to address the "Trustworthiness" aspect as opposed to reliability (essential for the quantitative research design).
> Page 8 of 15, Line 290: desist from self-citations for good scholarly practices. Instead, find cite other scholars within the discipline.
>I have no issues with your results and discussions.
> Make sure that your references are consistent, i.e., in references 13-14, you have used this ( (Vol. 24, No. 1):301–28), while in most regards (15, for example), you have used the shorter version (i.e., 92(5):717–28).
Author Response
Dear Reviewer,
we would like to thank you for the effort you took to evaluate our study. Your comments really helped to amend the quality of the paper. We have read all comments carefully and a detailed point-by-point response to your comments can be found on the following page. Additionally, we have highlighted those changes in the manuscript using track change.
>Section 2.3. Data analysis, line 164 (p. 5 of 15): This study adopts a qualitative research design; discuss how you are going to address the "Trustworthiness" aspect as opposed to reliability (essential for the quantitative research design).
- We are grateful that you highlight the importance of trustworthiness. We have highlighted the criteria for trustworthiness more clearly in the text and supported them with a reference (page 5, lines 177ff.).
The respective text passage reads as follows: „Several strategies were used to ensure the trustworthiness and quality criteria of the qualitative analysis, e. g. transferability, reflexivity, or credibility (42). A checklist was used to keep the interviews as consistent as possible and to ensure the transferability of the results. In addition, two forms of memos were used to analyse the interview data: an interview memo written immediately after the interview to reflect and document the setting of the interview, and evaluation memos containing the re-searcher's self-reflective thoughts during the interpretation process. Each step of the analysis was documented, e. g. the coding rules. The results were discussed within the research group. Additionally, the results were discussed with colleagues within a larger research project and preliminary results were presented at joint meetings to get feed-back from experts and participants. To increase credibility, the transcripts were sent back to the participants to check the content.“
(42) Korstjens, I., & Moser, A. (2018). Series: Practical guidance to qualitative research. Part 4: Trustworthiness and publishing. The European Journal of General Practice, 24(1), 120–124. https://doi.org/10.1080/13814788.2017.1375092
> Page 8 of 15, Line 290: desist from self-citations for good scholarly practices. Instead, find cite other scholars within the discipline.
- The aspect of the complexity of the technology was taken up by the participants in various facets during the interviews. In this paper, the aspect is presented only shortly as it is not the focus of the manuscript. However, the publication of Scheepers et al. (2022)—that we refer to in line 299—focuses on the aspects of software usability and complexity of technology mentioned by the same sample of interviewees. To allow interested readers to take a deeper look at the analyses on software usability and the complexity of technology, we found it necessary to keep the reference (page 8, line 312).
>I have no issues with your results and discussions.
- We are pleased that you appreciate our presentation of the results as well as the discussion.
> Make sure that your references are consistent, i.e., in references 13-14, you have used this ( (Vol. 24, No. 1):301–28), while in most regards (15, for example), you have used the shorter version (i.e., 92(5):717–28).
- Thank you for pointing this out. We have checked the references and revised them according to your suggestions (pages 15 + 16, lines ).
Sincerely,
Louisa Scheepers